# Stability of Genotube^®^ Swabs for African Swine Fever Virus Detection Using Loop-Mediated Isothermal (LAMP) Laboratory Testing on Samples Stored without Refrigeration

**DOI:** 10.3390/v16020263

**Published:** 2024-02-07

**Authors:** Dianne Phillips, Felisiano da Conceicao, Joanita Bendita da Costa Jong, Grant Rawlin, Peter Mee

**Affiliations:** 1Agriculture Victoria, Biosecurity and Agriculture Services, Bairnsdale, VIC 3857, Australia; 2Ministry of Agriculture, Livestock, Fisheries and Forestry, Government of Timor-Leste, Av. Nicolao Lobato, Comoro, Dili 0332, Timor-Leste; felisianodaconceicao@gmail.com (F.d.C.); lihirika@gmail.com (J.B.d.C.J.); 3Agriculture Victoria Research, AgriBio Centre for AgriBioscience, Bundoora, VIC 3083, Australia; grant.rawlin@agriculture.vic.gov.au (G.R.); peter.mee@agriculture.vic.gov.au (P.M.)

**Keywords:** ASFV, LAMP, Genotube swab, Timor-Leste, stability

## Abstract

African swine fever (ASF) is a transboundary viral disease which causes high mortality in pigs. In many low- and middle-income countries and in remote areas where diagnostic surveillance for ASF virus (ASFV) is undertaken, access to trained animal health technicians, sample collection, cold chain storage and transport of samples to suitably equipped laboratories can be limiting when traditional sampling and laboratory tests are used. Previously published studies have demonstrated that alternative sampling matrices such as swabs and filter papers can be tested using PCR without refrigeration for up to a week. This study used Genotube^®^ swabs stored in temperate and tropical climates without refrigeration for four weeks after collection to demonstrate there was no change in test performance and results using loop-mediated isothermal amplification (LAMP) ASFV detection on a series of pig serum samples including serum spiked with a synthetic ASFV positive control, naturally acquired ASFV positive serum from Timor-Leste and negative ASFV serum samples. The use of Genotube^®^ swabs for ASFV detection for surveillance purposes, coupled with testing platforms such as LAMP, can provide an alternative to traditional testing methodology where resources are limited and time from collection to testing of samples is prolonged.

## 1. Introduction

African swine fever (ASF) is a transboundary viral disease which causes outbreaks of significant morbidity and mortality in pig populations in countries around the world [1,2]. The number of regions and countries reporting ASF incursions has increased since 2005 to include Eastern Europe, the European Union, Asia, Oceania and most recently countries in the Caribbean [3]. The disease is usually associated with high mortalities in affected pigs, and commercially available vaccines are limited and still undergoing evaluation [1,2,4,5,6].

ASF virus (ASFV) infection of susceptible pigs results in viraemia and localization of the virus in various tissues. Laboratory tests for ASFV can be conducted on a range of sample types obtained from infected animals such as spleen, bone marrow, tonsils, blood, serum, faeces and saliva. These tests can be conducted on several platforms including virus isolation, haemadsorption, antigen detection by fluorescent antibody and detection of virus genome by polymerase chain reaction (PCR) techniques [1]. However, there are many situations, particularly in low- and middle-income countries (LMICs) where the availability of trained animal health technicians or veterinarians, sampling opportunities, cold chain storage and transport of samples and access to traditional laboratory facilities are limited. These resource limitations have stimulated research to investigate other sample collection methodologies and laboratory testing platforms to provide alternative techniques for ASFV detection, through either a variety of penside tests or sampling matrices which do not require cold chain storage [5,7,8,9,10,11,12,13,14,15,16,17,18,19,20,21,22].

Timor-Leste experienced its first outbreak of ASF in 2019 which resulted in a devastating loss of villager-owned pigs and major economic consequences to the country [23]. At that time, the national laboratory had no capability to test for ASFV. Over the course of the outbreak, in-country laboratory testing capacity was increased with the incorporation of firstly a loop-mediated isothermal amplification (LAMP) assay and, later, PCR diagnostics [10]. However, the collection of samples such as blood, swabs and tissues from pigs showing clinical signs of ASF for laboratory testing is still a challenging task. Remote location and difficult access to many pig farms, shortage of veterinarians and a lack of appropriate supply chain equipment, refrigeration and transport facilities are some of the major obstacles that hamper the collection and testing process (unpublished data and personal observation).

LAMP machines provide a highly sensitive and specific molecular laboratory test comparable with PCR but are portable, can be battery-operated and can provide test results in as little time as 30 min [18]. LAMP operations use selective primers to target specific viral sequences and polymerases to rapidly amplify the viral segments in a single cycle run at an isothermal temperature [24]. Positive amplification of the targeted viral sequences linked to fluorescent dye formation, followed by annealing of the amplified product at the expected temperature, provides confirmation of a positive sample. In the 2019 ASF outbreak in Timor-Leste, field verification of a ASFV LAMP assay was undertaken by these authors and showed comparable results for ASFV detection with qPCR test results [25]. 

Alternative sampling methods which do not require a high level of technical training for use and do not require refrigeration have previously been demonstrated to be suitable for ASFV diagnosis. Experiments using blood samples preserved on FTA cards and filter papers yielded equivalent results to EDTA blood samples using PCR testing [26]. Additionally, the filter papers still tested positive nine months later when stored at either temperate or high (37 °C) temperatures [27]. Dry blood swabs and commercially available quick-drying swabs (brands include Genotube, PrimeSwab and PrimeStore MTM) were also tested and found to be stable for testing for more than a week in laboratory conditions [14,15]. The commercially available quick-drying swabs have the advantage of simple operator use and impermeable storage suitable for transport along with preprinted labels/barcodes for ease of identification and tracking. 

This project investigates the extension of this principle to LAMP testing for ASFV in Timor-Leste in conjunction with Genotube^®^ swabs. Genotube^®^ swabs are a commercially available rapid-drying swab which can be used to collect a variety of sample types including blood, oral and faecal samples. Comparisons were performed across a series of treatments including Genotube^®^ swabs dipped in ASFV-negative sera (controls), ASFV-negative sera spiked with a synthetic ASFV positive and in naturally infected pig serum samples. Genotube^®^ serum swab samples were stored at ambient temperature in both a temperate and tropical climate and tested at weekly intervals for four weeks. The ASFV LAMP test results showed that the Genotubes^®^ swab successfully stabilised the samples and there was no change in test outcome (whether positive or negative) over the study period.

## 2. Materials and Methods

### 2.1. ASFV LAMP Detection

LAMP reactions were set up and performed in 8-well LAMP test trips using a Genie III (OptiGene, Horsham, UK) instrument with run conditions of 65 °C for 25 min, with annealing performed from 98 to 80 °C ramping at 0.05 °C per second. Reactions were set up as previously outlined in Mee et al., 2020 [10]. 

In general, individual swab samples were tested in wells 1–3, paired with the internal amplification control (IAC) dsDNA gBlock Gene Fragment (Integrated DNA Technologies) in wells 4–6 to check for evidence of inhibition of amplification of the sample and included a synthetic positive (dsDNA gBlock Gene Fragment (Integrated DNA Technologies)) and a negative control (nuclease-free water) in wells 7 and 8. Results from the Genie III LAMP tests are reported here as the time to peak ratio of fluoresence (PR) (minutes–seconds or equivalent seconds) and the anneal derivative temperature (Ta°C). Test results were analysed using Genie^®^ Explorer v2.0.6.3 software (OptiGene, Horsham, UK) using default thresholds.

LAMP results were called positive for ASFV if an amplified product had a Ta of 86.75 °C (±0.3 °C) and the PR was <20 min with a positive control result and no fluorescence or anneal product in the negative control well. Naturally acquired ASFV-positive and ASFV-negative serum samples were deemed not affected by inhibitors if the internal amplification control (IAC) gBlock was detected with a Ta of 89.5 °C (±0.5 °C) and a PR < 11 min. Samples which returned a PR value equal to or longer than 20 min and Ta within range were classified as indeterminate and retested.

### 2.2. Preparation of ASFV Synthetic DNA Spiked Genotube^®^ Swabs for Assessment of Stability over Four Weeks at Room Temperature (Five Replicates at Five Time Points) at Agriculture Victoria

Whole porcine (Sus scrofa domestica) blood was collected in plain blood tubes from a country free of ASFV. A low dilution of the synthetic positive control was prepared in nuclease-free water and spiked into porcine serum by adding 10 µL synthetic ASFV control into porcine serum (final concentration 20,000 copies/µL). A high-dilution synthetic positive serum sample was created by diluting the synthetic positive control with nuclease-free water and adding to a second aliquot of porcine serum (final concentration 2000 copies/µL). Preliminary LAMP testing of the low- and high-dilution serum samples gave PR values of 9 min 30 s and 14 min 45 s, respectively.

A total of 50 Genotube^®^ swabs were dipped into each of the low- and high-dilution serum samples to create 5 replicates for 5 time points. The swabs were allowed to dry overnight and were held at ambient room temperature for the entire testing period.

Five Genotube^®^ swabs, from each of the low- and high-dilution samples, were tested using the LAMP machine starting the day after swab preparation (Time 0) and at weekly intervals until four weeks had elapsed. Before testing, each swab was placed in 1 ml of nuclease-free water for a period of three minutes with intermittent stirring to elute serum off the swabs. An aliquot of 2 µL from each of the five replicates of low- and high-dilution samples was heat-treated in the Genie III machine for 95 °C for two minutes before using the ASFV LAMP assay. IAC wells were not used with these samples as they were prepared with a synthetic positive control and pretested to demonstrate that no inhibition of the LAMP reaction occurred.

### 2.3. Preparation of Naturally Acquired ASFV-Positive-Serum Genotube^®^ Swabs for Assessment of Stability over Four Weeks at Room Temperature (Dili)

A frozen serum sample collected from an infected pig from a previous ASF outbreak stored at the Dili Laboratory was retested using a Genie III machine to ensure the sample was still positive on LAMP.

A total of 25 Genotube^®^ swabs were dipped into the selected serum sample to create five replicates for five time points. The swabs were allowed to dry overnight and were held at room temperature for the entire testing period. 

Five of the Genotube^®^ swabs were tested using the LAMP machine starting the day after swab preparation (Time 0) and at weekly intervals until four weeks had elapsed. Before testing, each swab was placed in 500 µL of nuclease-free water for a period of 3 min with intermittent stirring to elute blood off the swabs. A smaller volume of nuclease-free water was used to maintain adequate concentration of viral DNA in the sample. An aliquot of 7.5 µL from each of the five replicates was heat-treated in the Genie III machine for 95 °C for 2 min before using the ASFV LAMP assay. Each sample well had a matching IAC well to demonstrate if there was any inhibition of the sample amplification. The volumes of LAMP master mix, primer mix and nuclease-free water were adjusted to reduce the dilution of the available viral DNA of the sample and keep the volume consistent at 27 µL for each test well (Table 1).

### 2.4. Preparation of ASFV-Negative-Serum Genotube^®^ Swabs for Assessment of Stability over Four Weeks at Room Temperature (Five Replicates at Five Time Points) at Agriculture Victoria

Whole porcine (Sus scrofa domestica) blood was collected in plain blood tubes from a Victorian pig abattoir. A total of 25 Genotube^®^ swabs were dipped into the serum sample to create five replicates for five time points. Swabs were tested as per methodology described in Section 2.2.

## 3. Results

All swabs created with synthetic ASFV-positive serum at both low and high dilutions tested positive with LAMP throughout the 4-week period (Table 2 and Table 3). The swabs were held at room temperature, which ranged from 20.1 to 25.9 °C (Melbourne, Australia).

Of the ASFV-positive serum replicates in Dili, 48/50 swabs tested positive on LAMP throughout the 4-week period. Two of the swabs tested at time zero had prolonged PR values when first tested, but Ta values were within the positive range. These samples were rerun and subsequently returned PR values within the positive range. Results for this series were tested on two runs of the LAMP machine due to the incorporation of IAC wells as per the Dili National Laboratory protocol (Table 4). There was some nonspecific fluorescence in the negative control wells on two test runs (Table 4, weeks 1 and 3) but no corresponding annealing product. Nontemplate amplification (nonspecific amplification and fluorescence in the absence of any template) has been reported in LAMP assays sporadically. It is thought to be more common in LAMP primers due to the additional bases and length compared with PCR primers [28,29]. In earlier research [10] comparing this ASFV LAMP with qPCR, there was also a small number of LAMP results with nonspecific fluorescence in sample wells, where matching qPCR results were consistently negative. In sample wells, this can be associated with the formation of “primer dimers” or as a result of contaminants in the sample. In the years of ASFV LAMP testing in Timor-Leste, both of these outcomes (nontemplate amplification and formation of primer dimers) have been rare (personal observation, unpublished data).

All Genotube^®^ swabs created with negative-ASFV serum tested negative throughout the 4-week period (Table 5). There was one replicate which generated a late PR value (week 4) with no corresponding anneal temperature which again indicates nonspecific fluorescence but not a positive test result. Daytime temperatures in the Dili Laboratory ranged from 24.5 to 31.3 °C. 

LAMP PRs from the naturally acquired ASFV-positive serum had the highest variance compared with results from the low and high dilution of the synthetic-serum swabs. This high variance reduced over time to be lower than the synthetic swab results by week 4 (Appendix A, Figure A1, Figure A2 and Figure A3).

## 4. Discussion

This study successfully demonstrated that Genotube^®^ swabs dipped in serum samples can be reliably tested for ASFV DNA using LAMP technology for up to a month from collection in both temperate and tropical climates without the need for refrigeration. 

Genotube^®^ swabs are advertised as stable without refrigeration, and a paper provided by the company demonstrated swabs of porcine oral fluids stored at room temperature produced stable real-time PCR (RT-PCR) test results at 1, 3 and 5 days, with a note that there was some deterioration in the viral signal observed at 5 days [30]. The paper contains an additional claim that nucleic acid contained in the swab will remain stable for years, but there does not appear to be data provided for this storage time frame. In published trials and research papers, Genotube^®^ swabs and other brands of commercial dry swabs were used for a variety of sample types from ASFV-infected pigs and stored for 3–8 days before testing on molecular platforms without impact on the outcome (positive or negative) of the test result [14,15,21]. Using qPCR tests for ASFV, these swab samples were found to give equivalent test results to traditional serum and/or EDTA blood samples. There was, however, a quantifiable difference in the qPCR Ct values in the range of roughly one log (about one Ct values) [14] and significant variation in the number of copies of viral genome detectable in a comparison of commercial swab types compared with EDTA blood samples [15]. The researchers noted that these effects were unlikely to change the outcome of test results when viral loads are high (usually with clinical cases). There are potential implications of reduced test sensitivity when viral loads are low such as preclinical or recovered cases [5]. 

Similarly in this trial, the variation in PR values for positive swab samples did not change the test outcome over the 4-week time period. Values at the higher end of the PR positive range were observed in the naturally acquired ASF-positive swabs compared with both the low- and high-dilution synthetic positive replicates. We hypothesise these higher values reflect the lower volume of serum available to create the swab replicates, which could result in more variable but generally lower quantities of viral DNA preserved on each swab. 

More broadly, various studies into extended storage of swab samples have recently been driven by the SARS-CoV-2 pandemic, and several studies demonstrated that storage of swabs in a variety of conditions including refrigeration and at ambient temperatures for up to 3 weeks did not affect the accuracy of test results using RT-PCR [31,32,33]. Although most research papers into sample stability use PCR diagnostic molecular platform, our trial looks at the use of LAMP, an alternative point-of-care portable test platform with high sensitivity and specificity which can provide an acceptable and comparable outcome [5]. In LMICs such as Timor-Leste, PCR-equipped laboratories are not always readily accessible, and a more flexible approach to diagnostic testing is required. 

Previous research on the ASFV LAMP test demonstrated that this test has a limit of detection of 4 × 10^2^ viral copies, with a PR value average of 19 min 50 s [10]. Our results suggest that if the amount of virus antigen DNA on the swab is within the limits of detection of the LAMP test, the effects of up to 4 weeks’ delay in testing does not affect the test outcome. Given the limited amount of stored ASFV-positive serum available for testing in Timor-Leste, further validation of naturally acquired ASFV-positive Genotube^®^ swab samples from ASF cases in early stages of infection or recovering cases where viral loads may be low would be beneficial. 

This paper provides evidence that extended time frames from the point of collection to laboratory testing of nonrefrigerated swab samples will not compromise test results for ASFV detection. Although timely diagnosis is usually critical to implementing ASF disease control measures, in many LMICs which are endemic for diseases such as ASF, disease outbreaks can be sporadic and isolated in area impacts. In Timor-Leste, there have been small and localised outbreaks of ASF since the initial epidemic in 2019–2020. According to unpublished surveillance data, there have been positive ASF virus test results on LAMP and/or PCR in these smaller outbreaks. There have also been other similar-sized isolated cases of pig sickness and/or mortality reports where ASFV or classical swine fever (CSF) was suspected but not confirmed. Whilst the outcome of these localised outbreaks often results in significant mortality of pigs for an individual small pig holder, there may be insufficient resources for a formal disease investigation and diagnosis. Barriers to disease investigation can include a lack of trained technicians or veterinary staff to visit the affected pig holder, reluctance by or ignorance of the pig owner to contact animal health staff or lack of sampling equipment, cold chain storage or suitable transport or diagnostic facilities [5,34,35]. This can lead to a frustrating lack of information about the causes of pig mortalities experienced by smallholder pig owners, especially when the clinical presentation is not pathognomonic for a specific disease and there are several endemic diseases which could be differential diagnoses [34].

The demonstration of the stability of the Genotube^®^ swabs for four weeks with no apparent loss in detection of ASFV DNA using a LAMP assay can add a valuable field tool for LMICs where gold standard laboratory tests may be limited and where the average time from sample collection to laboratory testing is variable and may be significantly extended from the ideal situation. The lack of cold chain storage and/or readily available transport to a diagnostic laboratory does not need to be a barrier to sample collection as this type of swab can still be accurately tested at one month and likely longer, providing valuable retrospective information about a pig’s ASFV status at the time of the sample collection.

## Figures and Tables

**Table 1 viruses-16-00263-t001:** Composition of LAMP sample wells for naturally acquired ASFV-positive swabs.

Constituent Name	Sample (Wells 1–3)	Sample and IAC (Wells 4–6)	Positive Control (Well 7)	Negative Control (Well 8)
LAMP ASFV master mix	15 µL	15 µL	15 µL	15 µL
10× primer mix	2.5 µL	2.5 µL	2.5 µL	2.5 µL
Template	7.5 µL	7.5 µL	0	0
IAC	0	2.0 µL	0	0
ASFV synthetic positive	0	0	2.0 µL	0
Water	2.0 µL	0	7.5 µL	9.5 µL
Total	27 µL	27 µL	27 µL	27 µL

**Table 2 viruses-16-00263-t002:** Genotube^®^ results of the porcine sera spiked with a low dilution of the synthetic positive control, Agriculture Victoria Laboratory September 2022.

Swab ID	Time 0 (25 August 2022)		Week 1 (1 September 2022)		Week 2 (8 September 2022)		Week 3 (15 September 2022)		Week 4 (22 September 2022)	
	PR	Ta	PR	Ta	PR	Ta	PR	Ta	PR	Ta
1	8:00	86.9	5:30	87.5	9:15	86.9	8:15	86.8	6:00	86.9
2	7:45	86.9	5:45	87.5	9:15	86.8	8:45	86.8	6:30	86.9
3	8:00	86.8	6:15	87.4	9:15	86.7	8:15	86.7	6:15	86.8
4	9:30	86.9	6:00	87.4	8:00	86.7	8:30	86.7	6:15	86.8
5	7:45	86.9	5:45	87.4	7:15	86.8	8:15	86.8	6:15	86.9
Positive control	11:00	87.2	3:45	87.5	3:45	87	4:30	87	3:45	87.1
Negative control	-	-	-	-	-	-	-	-	-	-

**Table 3 viruses-16-00263-t003:** Genotube^®^ results of the porcine sera spiked with a high dilution of the synthetic positive control, Agriculture Victoria Laboratory, September 2022.

Swab ID	Time 0 (25 August 2022)		Week 1 (1 September 2022)		Week 2 (8 September 2022)		Week 3 (15 September 2022)		Week 4 (22 September 2022)	
	PR	Ta	PR	Ta	PR	Ta	PR	Ta	PR	Ta
1	11:30	86.9	10:30	87.5	10:00	86.8	11:00	86.8	9:15	86.9
2	12:00	87.0	8:15	87.5	10:00	86.8	11:45	86.8	9:30	86.8
3	11:30	87.0	10:00	87.5	10:30	86.8	12:00	86.8	9:45	86.8
4	11:15	87.0	8:30	87.5	11:00	86.8	12:00	86.8	9:15	86.8
5	11:00	87.0	9:30	87.5	12:15	86.8	12:15	86.8	8:45	86.8
Positive control	4:15	87.0	4:00	87.6	4:15	87	4:30	87	3:45	87
Negative control	-	-	-	-	-	-	-	-	-	-

**Table 4 viruses-16-00263-t004:** Natural ASFV-positive results, Dili National Laboratory, March 2023.

Swab ID	Time 0 (11 March 2023)		Week 1 (17 March 2023)		Week 2 (24 March 2023)		Week 3 (31 March 2023)		Week 4 (7 April 2023)	
	PR	Ta	PR	Ta	PR	Ta	PR	Ta	PR	Ta
1	13:31	86.76	8:00	86.97	11:45	86.90	12:15	86.75	9:00	86.83
2	16:10	86.65	8:00	86.84	11:15	86:60	11:45	86.83	9:00	86.69
3	14:11	86.59	8:30	86.90	10:30	86.84	11:15	86.75	9:00	86.76
1 + IAC	6:32	89.30	4:30	89.44	6:00	89:29	5:15	89.36	5:15	89.29
2 + IAC	7:34	89.10	5:00	89.29	6:15	89:34	5:45	89.20	5:15	89.29
3 + IAC	6:52	89.01	5:30	89.29	6:15	89.38	5:15	89.30	5:30	89.34
Positive	5:12	86.65	5:30	86.83	5:45	86.87	4:30	86.93	8:45	86.18
Negative			2:00				24:15			
4	22:45	86.55	14:45	86.73	16:30	86.86	12:54	86.66	9:20	86.45
5	22:00	86.68	15:45	86.70	16:45	86.79	12:13	86.71	9:39	86.56
4 + IAC	9:00	89.31	8:30	89.30	9:30	89.28	6:51	89.15	5:43	89.21
5 + IAC	10:00	89.30	10:00	89.28	10:00	89.23	6:37	89.26	5:22	89.40
Positive control	6:15	86.88	7:45	86.84	9:00	86.88	5:56	86.88	5:06	86.61
Negative control	-	-	-	-	-	-	-	-	-	-

**Table 5 viruses-16-00263-t005:** ASFV-negative results, Agriculture Victoria Laboratory, October 2023.

Swab ID	Time 0 (6 October 2023)		Week 1 (13 October 2023)		Week 2 (20 October 2023)		Week 3 (27 October 2023)		Week 4 (3 October 2023)	
	PR	Ta	PR	Ta	PR	Ta	PR	Ta	PR	Ta
1	-	-	-	-	-	-	-	-	-	-
2	-	-	-	-	-	-	-	-	-	-
3	-	-	-	-	-	-	-	-	-	-
1 + IAC	4:00	89.89	4:45	89.90	4:00	89.89	4:00	89.90	6:00	89.90
2 + IAC	4:00	89.94	4:45	89.94	4:00	89.94	4:00	89.84	6:00	89.85
3 + IAC	4:00	89.99	4:45	89.89	4:00	89.99	4:00	89.90	6:00	89.90
Positive control	2:45	87.57	3:15	87.57	2:45	87.57	2:45	87.58	4:00	87.57
Negative control	-	-	-	-	-	-	-	-	-	-
4	-	-	-	-	-	-	-	-	-	-
5	-	-	-	-	-	-	-	-	20:00	-
4 + IAC	4:00	89.91	4:45	89.81	4:00	89.91	3:45	89.80	4:00	89.90
5 + IAC	4:15	89.80	5:00	89.71	4:15	89.80	3:45	89.70	3:45	89.89
Positive control	2:45	87.56	3:15	87.47	2:45	87.56	2:30	87.37	2:45	87.47
Negative control	-	-	-	-	-	-	-	-	-	-

## Data Availability

Data are contained within the article.

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
