# Peer review of "Stability of Genotube® Swabs for African Swine Fever Virus Detection Using Loop-Mediated Isothermal (LAMP) Laboratory Testing on Samples Stored without Refrigeration"

_viruses, 2024, doi:10.3390/v16020263_

Round 1

Reviewer 1 Report

Comments and Suggestions for Authors

I have minor comments to the technical part of the manuscript, but in general authors, in my understanding, make a wrong statement. I do not think that the described method allows to diagnose ASF.  Or we need to define the meaning of the word "diagnosis".

All diagnostic methods for ASF describe in WOAH Manual ASF.MANUAL2018 (woah.org) and LAMP is not included in this list. For example, if it will be a suspicion on ASF in Australia (or USA), is it will be acceptable to diagnose the ASF by recommended in this paper method? If not - probably we cannot call it "diagnosis".

Concerning the technical part:

- I did not find description of serum preparation. It is only mentioned that "Whole porcine (Sus scrofa domestica) blood was collected in plain blood tubes 172 from a Victorian pig abattoir. A total of 25 Genotube® swabs were dipped into 173 the serum sample to create five replicates for five time points. Swabs were tested 174 as per methodology described in section 2.2. 175".

Do the whole blood samples show the same  results? Is it feasible then you have lack of infrastructure obtain serum samples? How you plan to centrifuge blood in the field conditions?

Also the information about 2 false-positive results need to be described more in details, especially from the point of view how to avoid this situation in future.

"There 186 was some non-specific fluorescence in the negative control wells on two test runs 187 (Table 4, weeks 1 and 3) but no corresponding annealing product. This is occasionally 188 seen in LAMP testing associated with the formation of “primer dimers”. "

After this minor revisions, in my opinion this manuscript can be accepted for publication.

Author Response

Thank you for your review comments.

In response:

  • I agree that LAMP is not a WOAH recognised diagnostic test for ASFV and have amended the text to use ASFV detection. In Australia, we would use a LAMP test like this as a penside screening test but would follow up with PCR testing and other recognised tests for the final diagnosis.
  • The use of serum samples for the Genotube swabs reflects the difficulty of obtaining ASFV positive field samples in EDTA or plain blood tubes. Current field sample collection in Timor-Leste uses Genotube swabs usually dipped an ear vein blood sample. These samples can be transported without refrigeration on public transport. Blood tube samples require dedicated transport and equipment which is normally not available. Confirmed detections of ASFV in Timor-Leste over the last 2-3 years have been with Genotube samples submitted in this way, but which are not stored and hence not available for stability testing. The only samples we had for this trial were stored serum samples. In research planned this year we hope to collect some blood samples from the field from ASFV positive pigs, including some Genotube swabs with which we can repeat the storage and testing process.
        • In reference to the non-specific annealing products, after further reading I have recategorised this non-specific fluorescence as it occurred in the negative wells (no template) -see new paragraph and references in paper.

Reviewer 2 Report

Comments and Suggestions for Authors

The manuscript by Phillips et al. describes an assessment of Genotube swab samples for the detection of African swine fever virus DNA using LAMP assays. The results indicate the samples continue to generate positive results after storage at ambient temperatures for 4 weeks and thus may facilitate virus detection in resource-poor settings.

As I was reading this, I thought it would be useful if the LAMP assay was compared to the use of diagnostic real time quantitative PCRs and in true field situations but it was only in the Discussion section that it became apparent that the same group had published in 2020 a “field verification” of their system (ref [8]) but this is rather lost within a whole group of references [4-20] within the Introduction.  

I think the prior paper should be mentioned more explicitly in the Introduction so that the additional information provided in this new manuscript can be correctly assessed.

Most of the use of English is fine, as expected from an Australian lab, but there are a significant number of errors that should be corrected by careful reading by the authors. Some points are listed below.

Specific points:

1) In lines 219, 226 and 228 the abbreviation RT-PCR is used without explanation. I suspect the authors mean real time PCR since the assays are measuring viral DNA but in their previous paper (ref [8]) the authors used qPCR for this, which is better. RT-PCR is usually used for reverse transcription-PCR for the detection of RNAs.

Minor points:

a) Line 23, needs correction

b) Line 37, there are trials of ASFV vaccines in Vietnam. These have been given licences so the comment on line 37 should be modified.

c) Should it be Timor Leste or Timor-Leste? Both are used here, be consistent.

d) There is also inconsistency in the use of the abbreviations ASF and ASFV, one is for the disease and the other for the virus. The abbreviations should be used appropriately throughout the manuscript.

e) Line 152, text needs correction

f) It seems inappropriate to refer to “viral antigen” (line 163, line 241 and 257) when it is really the amount of viral DNA that matters.

f) Table 2 title (line 166)- “Constitution” seems an odd word to use here, please consider an alternative.

g) Line 245, “epidemic” could be replaced by “pandemic”

Comments on the Quality of English Language

As indicated above, some revisions are required (see lines 23 and 152) 

Author Response

Thankyou for your review suggestions. I have made the following changes:

  • Prior paper on ASFV LAMP test now explicitly explained in introduction.
  • In line 219, real time-PCR (expanded term) used (confirmed as test used in this reference) and in 226 and 228, qPCR inserted as test used in these linked references.
  • Line 23, corrected
  • Line 37, corrected and references relating to trials of ASFV vaccines in Vietnam inserted 
  • All mentions of Timor-Leste corrected to single format
  • Reviewed and amended use of ASF and ASFV
  • Line 152 corrected
  • References to “viral antigen” (line 163, line 241 and 257) amended to viral DNA 
  • Table 2 title (line 166)- “Constitution” amended to Components
  •  Line 245 “epidemic” replaced by “pandemic”